# Interfacial Peptides as Affinity Modulating Agents of Protein-Protein Interactions

**DOI:** 10.3390/biom12010106

**Published:** 2022-01-08

**Authors:** Pavel V. Ershov, Yuri V. Mezentsev, Alexis S. Ivanov

**Affiliations:** Institute of Biomedical Chemistry, 119121 Moscow, Russia; yu.mezentsev@gmail.com (Y.V.M.); alexei.ivanov@ibmc.msk.ru (A.S.I.)

**Keywords:** interfacial peptides, protein-protein interactions, inhibitors, preclinical studies, pharmacological targeting

## Abstract

The identification of disease-related protein-protein interactions (PPIs) creates objective conditions for their pharmacological modulation. The contact area (interfaces) of the vast majority of PPIs has some features, such as geometrical and biochemical complementarities, “hot spots”, as well as an extremely low mutation rate that give us key knowledge to influence these PPIs. Exogenous regulation of PPIs is aimed at both inhibiting the assembly and/or destabilization of protein complexes. Often, the design of such modulators is associated with some specific problems in targeted delivery, cell penetration and proteolytic stability, as well as selective binding to cellular targets. Recent progress in interfacial peptide design has been achieved in solving all these difficulties and has provided a good efficiency in preclinical models (in vitro and in vivo). The most promising peptide-containing therapeutic formulations are under investigation in clinical trials. In this review, we update the current state-of-the-art in the field of interfacial peptides as potent modulators of a number of disease-related PPIs. Over the past years, the scientific interest has been focused on following clinically significant heterodimeric PPIs MDM2/p53, PD-1/PD-L1, HIF/HIF, NRF2/KEAP1, RbAp48/MTA1, HSP90/CDC37, BIRC5/CRM1, BIRC5/XIAP, YAP/TAZ–TEAD, TWEAK/FN14, Bcl-2/Bax, YY1/AKT, CD40/CD40L and MINT2/APP.

## 1. Introduction

Stable and dynamic protein-protein interactions (PPIs) play extremely important roles in metabolic, regulatory and signaling processes in living systems [1,2,3]. The interactome shows the total spectra of all PPIs of an organism [4], which can be subdivided into tissue-specific, cellular, subcellular interactome and subinteractome of an individual protein [5]. Some PPIs may be involved in the formation of a pathological phenotype and are disease-associated. Therefore, the efforts of many research groups are aimed at both the identification of such PPIs and the search for ways of their pharmacological correction. Interactomes can be visualized as PPIs networks, which reflect physically and functionally interacting proteins [6,7,8]. As one rule, the central clusters in the network are represented by hub proteins with high connectivity, which are responsible for maintaining normal cellular processes. Therefore, a pharmacological modulation on at least one such hub is capable of causing a chain reaction of changes in the network topology and discoordination of a part of the interactome. However, this approach can be justified in the case of overexpression of proteins forming an aberrant disease-associated complex. To increase the pharmacological specificity of a PPI network, it is more appropriate to modulate peripheral clusters represented by molecular pathways containing functionally significant PPIs that are more abundant in pathology and predominantly absent in normal organisms [9,10]. Thus, PPI modulation is a promising innovative branch in fundamental and applied science with significance for translational and clinical medicine in the near future [11].

Three contrasting groups of PPI modulators can be distinguished, such as therapeutic proteins (Mr > 10 kDa), peptides (Mr = 1–5 kDa) and low molecular weight compounds of a non-peptide origin (Mr < 1 kDa) [12,13]. According to the mechanism of action, modulators are divided into allosteric and orthosteric. The former binds to sites on a protein surface, while the latter is directed to the contact area of two interacting proteins (interface) that form a homo- or heterodimeric complex. According to their biological effect, modulators are divided into those that interfere with complex assembly (inhibitors) and those that induce assembly (stabilizers) [14,15,16].

It is pertinent to note that there are a number of successful design solutions of effective PPI modulators based on low molecular weight non-peptide compounds [17,18,19]. Nevertheless, the scientific community is faced with a number of controversial questions addressed to the specificity of their action and sensitivity to disease-associated mutations in the interfaces of PPIs [20,21,22]. In this work, we focus on the analysis of the current investigations on peptide modulators of PPIs or interfacial peptides. Their development will facilitate the selection of new candidate drugs. The main leitmotif of this priority is due to the actual complexity of the development of drug-like low molecular weight compounds aimed at the protein-protein interfaces. For the overwhelming majority of protein complexes, the interface is a fairly extended surface with an area of more than 800 Å^2^. The stability of the PPI is provided through a set of non-covalent interactions in hot spots, that is, interactions of key amino acid residues (a.a.r) of both proteins, which account for the bulk of the total binding energy [12,23]. These hot spots are located relatively far from each other in the two-dimensional interface area, and hence, there is a need to select the compounds with a larger topological surface that, consequently, leads to an increase in molecular weight and deterioration of drug-like properties (especially, membrane permeability). On the other hand, there are difficulties in adapting the compound’s structure to achieve maximum geometric complementarity to several interface hot spots simultaneously [24]. The advantage of peptide modulators of PPIs is the ability to achieve maximum complementarity with lots of amino acid residues of one protein partner, thus competing for the binding site for the second protein partner. It should be noted that peptides positioned for solving therapeutic problems exhibit low toxicity due to their high affinity and binding specificity. However, traditional application limitations of classical linear peptides are in vivo instability, poor solubility and oral intake, low cell penetration capacity and immunogenic potential. The aim of this work was to update the current state of the art in the field of design, functional verification and the possibilities of clinical application of interfacial peptides as modulators of PPIs associated with pathologies. The scientific literature is regularly replenished with studies on new structural and biochemical features of PPIs’ interfaces; therefore, we will present the most important of them from the point of view of modulation by drugs.

## 2. Main Structural and Biochemical Features of PPIs’ Interfaces

The deciphering of interfaces of homo- and heterodimeric protein complexes is an empirical basis for the design of pharmacological modulators of PPIs. The contact area, which varies from 1000–4000 Å^2^ (average value 1600 Å^2^), is the main structural characteristic of the PPI interface. This range significantly exceeds the area of typical binding pockets with a more concave shape for the binding of low molecular weight metabolites or drugs. An area of such binding pockets ranges from 100–1000 Å^2^ [25,26]. It has been observed that protein interfaces with an area of less than 1200 Å^2^ are characteristic of shorter-lived and metastable complexes, while stable complexes have contact areas from 2000–4600 Å^2^ [27].

The amino acid composition of the contact area of two interacting proteins (complex “A–B”) provides molecular recognition specificity [28,29]. Conventionally, it is possible to distinguish mono- or multispecific PPIs when the contact area of protein “A” interacts either strictly with protein “B” or other protein partners, respectively. Specificity can be quantified by binding affinity, which describes the strength of the interaction between protein “A” and protein “B”. For the mono-specific complex “A–B”, the affinity is described by the equilibrium dissociation constant (K_D_), which means the concentration of free protein “B” at which half-saturation of protein “A” is achieved. Different scales of K_D_ values have been proposed, but the following is the most commonly used: low affinity (10^−3^ M K_D_ range), medium affinity (10^−9^ M K_D_ range) and high affinity (10^−12^ M K_D_ range) [29]. To visualize the interfaces and some features [30] of protein interactions and their relationship with affinity, Figure 1 shows the structures of three homo- and heterodimeric protein complexes as examples [31,32,33], which will also be discussed later in the context of inhibition of complexation by interfacial peptides.

It is generally accepted that the higher the affinity of a protein complex, the higher the likelihood that this complex has any functional significance. In most cases, this is true, although it is known that low-affinity short-lived protein complexes [34], including transient protein-peptide interactions in the cell, play important physiological roles [35]. On the other hand, aberrantly folded proteins that appear during a pathological process accompanied by a violation of the protein folding system can interact nonspecifically with other proteins with sufficiently high affinity due to an increase in the proportion of hydrophobic amino acid residues on the protein surface, which leads to the accumulation of protein aggregates [36].

The biochemical features of PPIs’ interfaces include complementarity and hydrophobicity [37]. The first one is defined as the condition for achieving the mutual positioning of surface amino acid residues of the corresponding interaction partners. In principle, complementarity can be chemical (formation of hydrogen, van der Waals and electrostatic bonds) [38,39] and geometric (shape) complementarity [40]. Thus, the presence of electrostatic complementarity between the surfaces of proteins allows the formation of salt bridges, which, in addition to hydrogen bonds and pi-pi stacking interactions between aromatic amino acid rings [41], stabilize the protein-protein complex and increase its lifetime. The complementarity of the contact regions of the two interacting proteins correlates with PPI’s affinity [42]. Accordingly, low geometric and chemical complementarity is expected to be associated with a labile complex formation, which is more vulnerable to modulating drugs. An interesting observation has been established in the case of complementarity of stable homodimeric complexes is much more pronounced than that of heterodimeric and short-lived complexes [43]. Hydrophobicity is associated with the number of water molecules in the contact area of proteins. An analysis of a sample of 3D models of crystallized protein complexes showed that the PPI’s interfaces are approximately 60% represented by nonpolar groups. Aromatic and hydrophobic amino acid residues form the interface core, while hydrophilic amino acid residues are located at the periphery of the core or within a globule [37].

## 3. Current Approaches to the Design of Interfacial Peptides

The design principles of interface peptides and the repertoire of technical solutions are detailed in recent reviews [44,45,46,47,48,49,50]. Optimization of the linear peptides is usually required to increase the binding affinity to the target protein, overcome a number of pharmacodynamic (pharmacokinetic) limitations and unfavorable physicochemical properties that reduce the in vivo efficacy. Table 1 lists design concepts of interfacial peptides, modulating the clinically significant PPIs [51,52,53,54,55,56,57,58,59,60,61,62,63,64], which will be discussed. The most common modification of a linear peptide is its cyclization by means of chemical stapling. This proceeds via cross-linking, that is, the addition of staples forming covalent bonds between amino acid residues located in different parts of a peptide or between the N- and C-terminus. Cyclization can also be conducted by oxidation of cysteine-containing peptides or in the presence of catalytic metals. Cyclization can significantly improve the pharmacological properties of the peptide due to the limitations of the conformational flexibility of the peptide chain and increase binding affinity, resistance to proteolytic degradation and cell internalization. Internalization of the peptide can be increased by conjugation with cell-penetrating peptides (CPPs), for example, GRKKRRQRRRPPQ, which is derived from the transactivating regulatory protein (Tat-protein) [65]. On the other hand, the ability of a linear or cyclic peptide to penetrate into cells is closely related to targeted delivery, which is especially important in the case of tumors or solid tissues. Affine peptides can be chemically cross-linked with specific motifs enriched in Arg-Gly-Asp or Asn-Gly-Arg, which ultimately allows exogenous peptides to efficiently penetrate into tumor cells. These motifs are recognized by cell surface proteins (integrins αvβ3 and αvβ5 as well as aminopeptidase N (CD13)) overexpressed in tumors [66].

Small molecule therapeutics can be fused with peptides that are sensitive to changes in the pH of the medium. This makes targeted delivery of peptides to tissue sites possible within a slightly more acid milieu (less than pH 7.0), which is the characteristic of tumors and/or their microenvironment [67] as well as inflamed tissues [68]. Under slight acid conditions, a pH-sensitive peptide takes the conformation of the alpha-helix that favors its incorporation into the cell membrane [69,70]. Srivastava and co-authors demonstrated that the charge regulation mechanism is the most important contributor in protein-polyelectrolyte complexation regardless of pH and other physical chemistry parameters using constant-pH Monte-Carlo simulations [71]. In addition, it has also been recently shown that the presence of “charged patches” along the polyelectrolyte is also an important factor in the stability of the protein/peptide interactions [72,73].

## 4. Targeting Clinically Significant PPIs with Interfacial Peptides

We used the Scopus literature database to search for publication statistics and citations (2017–2021) in the design of biologically active small molecules that target the interface of protein complexes. We found that the greatest scientific interest (the number of scientific studies and the level of their citation) in the field over the past 5 years has been observed in the design of PPI modulators for the following pairs: PD-1/PD-L1, MDM2/p53 NRF2/KEAP1 (Figure 2). Another group of PPIs (HSP90/CDC37, CD40/CD40L, YAP/TAZ–TEAD, TWEAK/FN14, HIF-1/HIF-1) is characterized by a much smaller number of publications. Using Google Scholar, we further identified at least eight PPIs that have emerged over the past two years in the context of the design of peptide inhibitors targeting the interface of two identical (NFAT5/NFAT5 and TYMS/TYMS) and two different protein subunits (BIRC5/CRM1, BIRC5/XIAP, Bcl-2/Bax, RbAp48/MTA1, YY1/AKT). It can also be focused on cancer-associated proteins such as LDH5 and SRPK1. Unlike LDH5, protein kinase SRPK1 is monomeric and phosphorylates a number of protein substrates with RS domains. Thus, the contact area of stable protein complexes with the participation of modifying enzymes is another molecular target of interfacial peptides.

Most studies devoted to the design of interfacial peptides, as a rule, include several sequential stages: in silico simulations, peptide synthesis, quality assessment and experimental verification. The directional design of an orthosteric peptide modulator depends on the completeness of structural information about the contact area of a targeted binary PPI, which is possible via a 3D crystallographic model of each of the proteins, or better, the protein complex itself. Protein-peptide interactions are initially assessed through binding kinetic and affinity constants measurement or in biochemical tests. Cell and animal models of disease are also used for the assessment of phenotypic effects induced by interfacial peptides that block the signaling or metabolic pathways in which the targeted PPI is involved. Below, we will discuss the results of studies, mainly, over the past 2 years, in the field of identification, design and inhibitory activity of interfacial peptides.

### 4.1. Targeting Homo-Oligomeric Protein-Protein Interactions with Interfacial Peptides

#### 4.1.1. LDH5

Lactate dehydrogenase 5 (LDH5) consists of four type-A subunits and is overexpressed in metastatic tumors, making LDH5 an attractive target for pharmacotherapy. Using in silico simulations (FF14SB force field Amber 20 package and Rosetta), a peptide with a high affinity to the beta-sheet region of the LDH5 enzyme oligomerization interface was constructed. In order to increase the stability and potency, it was integrated into a beta-hairpin peptide scaffold, rendering good penetration into the cells. This variant inhibited LDH5 activity in vitro much more strongly than the low molecular weight compound GNE-140 (Mr = 499) used as a known reference [74].

#### 4.1.2. TYMS

The inhibition of thymidylate synthase (TYMS, EC 2.1.1.45) is used in anticancer chemotherapy. This homodimeric enzyme converts deoxyuridine monophosphate (dUMP) to deoxythymidine monophosphate (dTMP). A conjugate of TYMS-derived affinity peptide (LSCQLYQR) and folic acid was constructed. The conjugate enters cells through folate-binding protein, which is overexpressed on the plasma membrane of tumor cells and targets the contact area of two TYMS subunits shifting the monomer-dimer equilibrium towards the formation of inactive monomers. Experiments showed the specificity of delivery of the conjugate to tumor cells and the suppression of cell growth with no TYMS de novo expression. [75].

#### 4.1.3. NFAT5

The transcriptional regulator, nuclear factor of activated T-cells 5 (NFAT5), is still the least studied among other factors of the NFAT family. It is known to regulate the expression of genes associated with autoimmune diseases and diabetes mellitus [76,77]. One of the peptide inhibitors was capable of targeting NFAT5 through the C-terminal dimerization and DNA binding domain and was first demonstrated in silico. Molecular dynamics and mechanics with the solution of the Poisson–Boltzmann surface area (MM/PBSA) equation were used to select a number of interfacial peptides with the highest affinity for NFAT5 [78]. Unlike other calcineurin-dependent NFAT factors, NFAT5 is involved in thymocyte maturation, survival and T-cell proliferation, so the benefit of specific inhibition of NFAT5 for the treatment of autoimmune diseases remains controversial [79].

### 4.2. Targeting Hetero-Oligomeric Protein-Protein Interactions

#### 4.2.1. MDM2/p53

The driver of numerous studies devoted to the modulation of PPIs between the tumor suppressor p53 with its negative regulators MDM2 (mouse double minute 2 homolog, E3-ubiquitin-protein ligase) and MDMX (p53-binding protein Mdm4) is its proven clinical significance [80]. In addition, the MDM2/p53 complex, as well as PD-1/PD-L1 (see Section 4.2.2), can be legitimately regarded as structurally well-characterized reference molecular models for the design and demonstration of innovative approaches for targeting PPI interfaces by various pharmacologically active agents, including those of peptide origin. Meanwhile, most of the studies were aimed not so much at identifying new affine interfacial peptides but rather at potentiating the capability to penetrate into cells. Thus, the PMI-M3 peptide (LTFLEYWAQLMQ) was identified, which interacted with the p53-binding pocket of MDM2 and MDMX (K_D_ in pM range). To increase the penetration into cells, the peptide was modified by attaching lysine residues to the C- and N-terminus, which made it possible to enhance its p53-dependent anticancer activity [81]. On the basis of another peptide inhibitor, KD3, conjugates with a cell-penetrating peptide (cTAT-KD3 and cR10-KD3) were designed, which were able to activate the p53-dependent signaling pathway in cell models when exposed to a lower micromolar range [82]. Peptidomimetics with extremely high resistance to proteolysis in vivo based on right-handed alpha-helical sulfono-γ-AA peptides inhibited the complex formation of MDM2/p53 and MDMX/p53 by binding to MDM2 and MDMX with K_D_ values of 19 and 67 nM, respectively. Circular dichroism, 2D-NMR spectroscopy and in silico modeling have shown that these molecules mimic the p53 amino acid motif [83]. The peptide with amino acid sequence TSFAEYWALLSP, which binds to MDM2 and MDMX with K_D_ values of 0.49 and 2.4 nM, respectively, was conjugated to gold nanoparticles, which, by analogy with CPP, help to overcome low cell permeability. It is interesting to note that this construct had much higher in vitro activity than the reference low molecular weight compound Nutlin-3 (Mr = 582, CAS No, 890090-75-2) [84]. It should be noted that studies [81,82,83,84] were focused more on achieving high binding affinity of interfacial peptides and optimizing cell penetration, but not on achieving specificity of delivery to tumor cells. However, a study by Lundsten and co-authors [85] addressed this problem. The authors showed that a tumor-specific nanocarrier, based on an epidermal growth factor receptor (EGFR)-targeted lipid bilayer disks, efficiently delivered p53-activating stapled peptide VIP116 to EGFR-expressing tumor cells and decreased tumor cell viability.

We compared median values of interfacial parameters of crystallographic complexes (*n* = 30) from Protein Data Bank (PDB) consisting of MDM2 (MDMX) protein and peptides or low molecular weight compounds (Table 2) by using the PDBePISA v1.52 web-based tool. These inhibitory agents targeted the contact area of MDM2 and p53, thus preventing protein-protein interaction. It can be seen that due to almost two higher values of the solvent-accessible surface area of peptide PPI inhibitors, a larger interface area forms with three times more hydrogen bonds. Since the different methods and experimental conditions (for example, temperature) for K_D_ determination of complexes were used, it is hard to perform a comparative analysis between the two groups.

#### 4.2.2. PD-1/PD-L1

The interaction of the PD-1 (programmed cell death-1 receptor) for its ligand PD-L1 allows tumor cells to escape immune surveillance. The paradigm based on the inhibition of immune checkpoints PD-1/PD-L1 has been approved as a standard of treatment for non-small cell lung cancer (NSCLC) [86] and for other neoplasms [87,88]. In the last decade, several anti-PD-1/PD-L1 monoclonal antibodies have been approved, and an increasing number of peptide and non-peptide small molecule inhibitors have been discovered. Indeed, the functional characterization of several leader peptide structures interfering with PD-1/PD-L1 interaction in in vitro and in vivo models has been reported [52,89,90,91]. Figure 3 shows a model of the PD-L1 complex with an interface peptide, which hinders binding with PD-1 therapeutic peptides, showing that low toxicity in vivo with concomitant optimization of targeted delivery to the tumor microenvironment [92] can be considered as candidates for cancer immunotherapy. Thus, a peptide derived from the peroxiredoxin-5 motif inhibited the interaction of PD-1/PD-L1 with an IC50 of 0.6 μM and tumor growth [93]. Interesting data were obtained in the study by Liu and co-authors [94], where the anti-PD-1/PD-L1 interface peptide also modulated the CD80/PD-L1 interaction. In comparison with antibody preparations, it showed a higher efficiency of penetration into 3D models of tumor spheroids, increasing the immune response of tumor-infiltrating T cells (co-cultured with cancer cells) and preventing their apoptosis. The PPI interfaces predominantly represented by beta-sheets can be targeted by peptides with beta-hairpin structures. The peptide extracted from the PD-1 sequence was the basis for the peptide-dendrimer conjugate, a branched polymer whose “branches” originate from a single center. This design allowed it to stabilize peptide structure and increase the binding affinity for PD-L1 by almost five orders of magnitude compared to the linear peptide, presumably due to multivalent binding [95]. Using the method of bio-layer interferometry and cell models, it was shown that peptide inhibitors based on the PD-L1 motif consisting of 14 amino acid residues were able to form a stable hairpin structure. They blocked PD-1/PD-L1 complex formation through a high-affinity interaction with PD-1, which was accompanied by an increase in IL-2 secretion [96]. Within the logic of similar ideology and methodology, the peptidomimetic PL120131, which inhibited the PD-1-mediated apoptotic signaling pathway and recovered Jurkat cells (a model of acute T-cell leukemia) as well as primary lymphocytes from apoptosis, was designed [97].

It is important to mention the prospects for the development of peptide-based drugs with exceptional resistance to proteolysis for cancer immunotherapy with oral administration. The PD-1/PD-L1 blocking peptide was co-administered with *N*, *N*, *N*-trimethyl chitosan, a hydrogel increasing the bioavailability of the peptide by 53% in mouse models. This made it possible to achieve not only inhibition of tumor growth but also an increase in tumor infiltration by CD8+ T cells and secretion of IFN-γ [99].

Photothermal ablation in combination with a PD-1 pharmacological blockade can effectively kill primary tumors and suppress the growth of secondary tumors [100]. Hollow gold nanoshells (photothermal agent) and an interfacial peptide to PD-1 were co-encapsulated into nanoparticles of a biodegradable copolymer. It has been shown that the intratumoral administration of the composition is capable of releasing the peptide for up to 40 days. After exposing the tumor site to near-infrared laser radiation, a positive effect of killing tumor cells was achieved [100].

#### 4.2.3. HIF-1

The hypoxia-inducible factor HIF-1 is a master regulator that is critical for tumor metabolism. HIF-1 forms different functional significant heterodimers. As an example, it can be considered the heterodimerization of HIF-1α and HIF-1β, which is important for the proper binding to DNA regions [101]. It should be noted that drug targeting of HIFs is still a difficult task. However, screening the cyclic hexapeptide library revealed cyclo-CLLFVY, which inhibited HIF-1α/HIF-1β dimerization by binding to the PAS-B domain of HIF-1α (K_D_ ≈ 124 nM) [102]. Another example is a heterodimeric high-affinity HIF-1α complex with the p300/CBP transcriptional co-activator (K_D_ ≈ 7 nM) [103], and after structure optimization of the interface peptides, certain positive results were achieved in targeting this PPI [104].

#### 4.2.4. NRF2/KEAP1

The disruption of the complex formation between transcription factor NRF2 and KEAP1 (Kelch-like ECH-associated protein 1) has a high therapeutic potential for a number of pathologies accompanied by oxidative stress (neurodegenerative, chronic obstructive pulmonary and inflammatory diseases). Interfacial peptides against NRF2/KEAP1 showed a high binding capacity [105] but had insufficiently high activity in cells so far; therefore, the design of modified peptide structures has launched a new round of studies.

A new promising cyclic interfacial peptide, based on the principle of "head-to-tail" containing minimum acidic amino acids and Gly as a linker for C- and N-terminus, has been developed. Cyclization eliminated the charge of the terminal residue and stabilized the conformation of the peptide upon binding to KEAP1. The peptide, as shown by experimental verification, promoted the activation of NRF2 at the cellular level [106].

Another work [107] reports that due to the restriction of conformational flexibility through the disulfide and perfluoroalkyl bridges in the peptide structure mimicking the beta-turn of KEAP1, it was possible to inhibit Nrf2/Keap1 interaction. Peptides with cyclic structures have shown good efficacy in modulating this PPI when conjugated with CPP. Such dipeptides not only retained the binding with KEAP1 but were also resistant to proteolytic degradation and easily penetrated cells, inducing the transcriptional activity of NRF2 [108].

The peptide modulator of NRF2/KEAP1 based on the NRF2 motif was modified by the introduction of a disulfide bond, then conjugated with CPP and grafted into MCoTI-II cyclotide scaffold. Further experimental verification showed (i) high binding affinity for KEAP1; (ii) up-regulation of NRF2-dependent genes NQO1 and TALDO1; (iii) proteolytic stability in serum [109]. It is necessary to clarify that cyclotide is a microprotein of plant origin containing from 30 to 40 amino acid residues. It is cross-linked by three disulfide bonds forming a so-called cystine knot, which provides cyclotides exceptional thermal and chemical stability, as well as resistance to proteolytic degradation [110].

#### 4.2.5. RbAp48/MTA1

Retinoblastoma-associated protein 48 (RbAp48, RBBP4 gene), being a component of several histone-modifying complexes (Hat1, NuRD, PRC2, and CAF-1), is overexpressed in some tumors and, in turn, forms a functionally significant complex with the MTA1 (metastasis-associated protein 1). Based on the linear amino acid motif of MTA1, a cyclic peptide was created. It inhibited the RbAp48/MTA1 through interaction with the contact area of RbAp48 in the low nanomolar range (K_D_ ≈ 9 nM) and was markedly resistant to proteolytic degradation [111].

#### 4.2.6. Interaction of SRPK1 with Its Protein Substrates

Ser/Arg protein kinase 1 (SRPK1) carries out post-translational regulation of the splicing factor SRSF1 via phosphorylation of its C-terminus enriched in Ser and Arg. SRSF1 influences the synthesis of a pro- or anti-angiogenic splice variant of vascular endothelial growth factor (VEGF). The peptide-based inhibitor (20 amino acid residues) had excellent cell penetration and blocked phosphorylation as well as binding of protein substrates to SRPK1 and thus was able to switch splicing from pro-angiogenic to anti-angiogenic VEGF variant [112].

#### 4.2.7. HSP90/CDC37

Heat shock protein 90 (HSP90) and its co-chaperones are overexpressed in tumor cells and involved in neoplastic transformation [113]. Therefore, pharmacological destabilization of PPIs in this system will provide advantages in selective inactivation of “client” proteins. Close attention was paid to the development of interfacial peptides modulating the protein complexes HSP90/CDC37 and HSP90/CDK4/CDC37, in which HSP90 and CDC37 perform the correction of “client” protein maturation, for example, cyclin-dependent kinase 4 (CDK4). Five interfacial peptides were selected on the basis of CDC37 motifs. It was shown that only two of them, Cdc37p3 (NYSVWDHIEVEDDLSKDGFSKSMV) and Cdc37p5 (PSKDIFLKSMIN), inhibited the maturation of CDK4 in the low micromolar range, and there was no effect for CDC37-independent protein kinases [114]. Two other peptides, Pep-1 (Ac-KHFGMLRRWDD-NH2) and Pep-5 (Ac-HFGMLRR-NH2), blocked HSP90/CDC37 complex formation in vitro through interacting with the N-terminal domain of HSP90 with K_D_ ~ 7 μM and ~6 μM, respectively. Pep-1, in addition, inhibited the ATPase activity of HSP90 (IC50~3.0 μM) [115,116].

#### 4.2.8. BIRC5/CRM1

BIRC5, baculovirus inhibitor of apoptosis repeat-containing 5 or survivin, functions as a mitotic and apoptotic regulator in the chromosomal passenger complex (CPC). Exportin-1 (CRM1) is critical for CPC binding to centromeres when interacting with BIRC5, and thus, modulating the heterodimeric BIRC5/CRM1 complex, a number of pathogenic processes, including tumor cell proliferation, can be affected [117]. The artificial molecular tweezers consist in the design of open macromolecules capable of non-covalently capturing client molecules or certain amino acids such as lysine and arginine with high affinity. They have an electron-saturated torus-shaped cavity containing two phosphonate groups. Exceptional selectivity for these two amino acids is achieved by passing the entire amino acid side chain through the cavity and then locking by a phosphonate-ammonium salt bridge [118]. Conjugates of lysine-selective supramolecular tweezers with covalently linked peptides _95_ELTL_98_ and _95_ELTLGEFL_102_ based on the motifs corresponding to the BIRC5 interface area were designed [119]. In the absence of protein partners, BIRC5 monomer-dimer equilibrium is shifted towards the dimer. The contact area between monomers (_93_FEELTLGEFL_102_) is self-complementary and overlaps with the sequence _89_VKKQFEELTL_98_ related to the nuclear export signal (NES). The lysine residue at position 103 located at the very beginning of the C-terminal of BIRC5 has been suggested as the most suitable “anchor” for capture with supramolecular tweezers. Lysine residues 90 and 91 were chosen as alternatives [119].

#### 4.2.9. BIRC5/XIAP

The interaction between BIRC5 and XIAP (X-linked inhibitor of apoptosis protein) confers XIAP stability to ubiquitin-dependent proteasome degradation and promotes synergistic inhibition of apoptosis, which is absent in XIAP −/− deficient cells [120]. Therefore, the design of peptides that target the BIRC5/XIAP interface is important in the context of activating anti-tumor mechanisms in cells. Peptide Sur-X, with the sequence derived from the XIAP-binding region (K15-M38) of survivin and the cell-penetrating sequence from HIV Tat protein added to its N-terminal, was shown to destabilize the BIRC5/XIAP complex and induced growth inhibition and necroptosis in HCT116, HCT15, RKO, and HT29 cells and also had a pro-apoptotic effect in vivo in mouse xenograft models [121].

#### 4.2.10. YAP/TAZ–TEAD

The Hippo signaling pathway can take part in neoplastic transformation by means of PPI between the oncoprotein YAP (yes-associated protein) and TAZ (transcriptional co-activator with PDZ-binding motif) and TEAD (transcription factor). The formation of YAP/TAZ-TEAD complexes is important for the transcription of growth-promoting genes. Crystallographic data on YAP/TAZ-TEAD complex allowed the design of linear interfacial peptides with an irregular structure (omega loop) containing the maximum number of “hot” amino acid residues critical for PPI. The peptides were cyclized through disulfide groups attached to C- and N-terminus. Experimental verification confirmed that the cyclic peptides YSP-2 (_70_PMRLRKLPDSFFK_82_) and TSP-2 (_42_SWRKKILPESFFK_54_) had 3.7–6.6 times higher binding affinity with TEAD protein. In silico models showed the reduction in free energy of protein-peptide complexes through the entropy term [122].

In addition to interfacial peptides, the number of low molecular weight compounds aimed at inhibiting YAP/TAZ-TEAD complex formation, but only two compounds showed moderate blockade of the Hippo pathway [123]. More recently, it was found that the compound NSC682769, a benzazepine derivative, at submicromolar concentrations inhibited YAP/TEAD1 (TEA domain family member 1) interaction in glioblastoma cells [124].

#### 4.2.11. TWEAK/FN14

Fibroblast growth factor-inducible 14 (FN14) is a well-known membrane receptor for cytokines, in particular, TWEAK (tumor necrosis factor-like weak inducer of apoptosis). Both proteins are overexpressed in many carcinomas, metastatic foci and associated with poor survival prognosis. The selection of 50 interfacial peptides that mimic motifs in the contact area of proteins was carried out using computer modeling. The first round of experimental verification consisted of peptide treatment of TWEAK-dependent and independent cell cultures, followed by transcriptome analysis of a panel of nine downstream genes in TWEAK/FN14-mediated signaling pathways. For those four peptides that suppressed the expression of these genes, an additional surface plasmon resonance verification of peptides binding to TWEAK was performed [125].

#### 4.2.12. Bcl-2/Bax

The apoptosis regulator Bcl-2 (B-cell lymphoma 2) suppresses the pro-apoptotic functions of Bax (Bcl-2-like protein 4). The BH3-domain containing proteins activates Bax, competing with Bcl-2, thereby opening up the possibility of creating peptidomimetics based on BH3 motifs. The limitations of the pilot variants of peptidomimetics were their low level of cell penetration and proteolytic degradation, which were further overcome by conjugation with spheroidal gold nanoclusters. At the same time, polymer elements (Au1^+^-S-BH3]n were capable of self-assembly in situ on the surface of gold nanoparticles by means of one-pot synthesis [126].

#### 4.2.13. YY1/AKT

The oncoprotein binding domain of YY1 (Yin Yang 1) located between Gly201 and Ser226 is involved in the regulation of cell proliferation through PPIs with MDM2, EZH2 E1A, and AKT (RAC-alpha Ser/Thr protein kinase). A peptidomimetic corresponding to Gly206-Ser226 motif inhibited the YY1/AKT complex, proliferation and migration of tumor cells. In addition, the peptide caused the arrest of the cell cycle in the G1 phase and a decrease in the level of phosphorylation of Ser473 of AKT [127].

#### 4.2.14. CD40/CD40L

The KGYY-15 peptide was derived from the amino acid motif CD40L, which is responsible for the interaction with CD40 (tumor necrosis factor receptor superfamily member 5). KGYY-15 showed a weak potential for inhibiting the CD40/CD40L interaction (IC50 = 202 μM, 95% CI 144–288 μM). Verification of the peptide in the cell tests revealed only insignificant inhibitory activity (~33%) at a concentration of 100 μM. It was also known that low molecular weight compound DRI-C21095 disrupted the interaction of CD40 with other protein CD154 (IC50 = 9 μM) [128,129].

### 4.3. Inhibitors of Protein Polymerization

The abnormal accumulation of beta-amyloid (Aβ) accompanies Alzheimer’s disease. Aβ is formed during the proteolytic cleavage of APP (amyloid precursor protein). The hypothesis that pharmacological targeting of PPI Mint2/APP can influence the processing and formation of pathogenic Aβ was verified in [130]. It was shown that the effect of a peptide inhibitor on Mint2/APP correlated with a decrease in the accumulation of pathogenic variant Aβ42 in vitro. On the other hand, carcinogenesis is known to be inversely related to Alzheimer’s disease. At the molecular level, this is manifested in the inhibition of tubulin polymerization by Aβ peptides and apoptosis-dependent tumor cell death. Computer modeling predicted Aβ binding modes near the vinblastine-binding site of tubulin (H6-H7 loop). Then, based on the sequence of Aβ, peptides P1 (FRHYHHFFELV) and P9 (HYHHF) were designed to inhibit tubulin polymerization, the mechanism of action of which may be mediated through disruption of nucleotide metabolism [131]; however, the efficacy of these peptides is yet to be tested.

### 4.4. Targeting PPIs Involving Viral and Human Proteins

Retrospectively, the efficacy of interfacial peptides has been extensively studied in relation to inhibiting HIV1 protease homodimerization and reverse transcriptase heterodimerization in the late 1990s and early 2000s [132,133]. Currently, the main focus is on the design of peptidomimetics aimed at the active site of enzymes [134,135]. One of the approaches to drug design against SARS-CoV-2 is associated with the interference of viral Spike-protein and angiotensin-converting enzyme 2 (ACE2) interaction with peptide-based agents [136,137,138]. The work of Maas and co-authors demonstrated that the cross-linked peptidomimetics hACE221-55A36K-F40E, hACE221-55F32K-A36E and hACE221-55F28K-F32E inhibited RBD/ACE2 complex formation with IC50 and K_D_ values in the low micromolar range [139]. Oncoprotein E6 of the human papillomavirus (HPV) plays an important role in maintaining malignancy and affects the oncogenic phenotype through PPIs with more than a hundred proteins of human cells. Thus, drug modulation of cross-species PPIs could be one of the therapy applications for HPV-associated tumors [140]. However, so far, such interactions as E6/p53 and E6/CASP8 are considered mainly as targets for low molecular weight non-peptide affinity and functional state modulators of these protein complexes.

## 5. Conclusions

For a panoramic view of the relationships between the cellular and pharmacological context, we performed a functional enrichment of a set of target proteins with the categories “gene ontology”, “human phenotype”, “pathway”, “drug” and “disease” using the ToppFun tool of the ToppGene Suite annotation platform [141]. Table 3 shows the main subcategories containing the maximum number of target proteins. It follows from Table 3 that more than half of the target proteins in the gene-disease-drug axis are associated with the processes of neoplastic transformation, which, in principle, is logical since this scientific field is one of the most important in bio- and translational medicine. A strategy of pharmacological targeting conservative contact regions of molecular complexes with the participation of intensively studied PD-1, PD-L1, p53, Mdm2, Mdmx, KEAP1 and NRF2 proteins extrapolates to a group of homo-oligomeric metabolic enzymes, for example, TYMS and LDHA, which functions in homodimeric and homotetrameric forms, respectively, and overexpresses in some tumors (Figure 4). 

As for the current section of recruited and launched clinical trials (CTs) of therapeutic peptides, their number is still quite modest. It is in marked contrast with the total number of clinically significant PPIs whose interface can be modulated by drugs. In the analytical review by Cabri and co-authors [142], there were found 58 therapeutic peptides that occurred in CT’s reports. Of these, 13, 26 and 15 are in phase I, II and III, respectively, and 4 peptides are close to FDA approval (BBT-401, Foxy-5, Nangibotide and Reltecimod) [142]. The first trial with 71 enrolled participants studied ALRN-6924 formulation containing the peptide inhibitor of MDM2/p53 for tumors with the wild-type TP53 gene. ALRN-6924 demonstrated dose-dependent pharmacokinetics and increased levels of serum MIC-1 protein, a biomarker of the activation of the p53-dependent signaling pathway [143].

Targeting the interfaces of disease-associated PPIs with peptide modulators is a successful strategy for correcting some pathological processes. The optimization of design approaches, pharmacological and binding kinetics profiles of peptide inhibitors will contribute to the development of highly selective and affine drug prototypes. In turn, the progress in interfacial peptides is organically linked with the identification of novel clinically significant PPIs, advances in deciphering 3D crystallographic models of protein complexes and mapping hot amino acid residues in the contact area of proteins. Along with peptide modulators of PPIs, there is much literature data on pharmacologically active low molecular weight organic compounds and therapeutic protein. All the above-mentioned facts directly testify to the growing interest in this field from the scientific audience.

Recently, the AlphaFold2 approach has emerged as a powerful in silico instrument for the prediction of structure, topology, conformation and variants interpretation of most soluble and membrane proteins [144,145]. In the context of drug discovery, it will contribute to the optimization of binding models of pharmacological agents to their novel target proteins. However, predicted models of molecular complexes are still structural hypotheses requiring experimental verification.

## Figures and Tables

**Figure 1 biomolecules-12-00106-f001:**
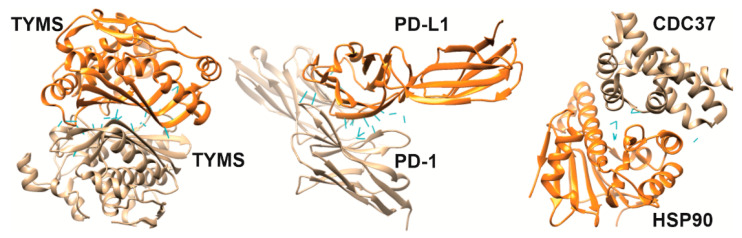
Binding interfaces of protein complexes based on crystallographic data (X-ray diffraction) and the main parameters of the interfaces calculated using PDBePISA [30]. Homodimer thymidylate synthase (TYMS) (PDB ID: 1HZW): interface area—2260 Å^2^, hydrogen bonds (HB) = 27, salt bridges (SB) = 2, K_D_ = 200 nM [31]. PD-1/PD-L1 (PDB ID: 3BIK): interface area—863 Å^2^; HB = 19, SB = 5, K_D_ = 8 µM [32]. HSP90/CDC37 (PDB ID: 2K5B): interface area—721 Å^2^, HB = 7, SB = 8, K_D_ = 100 µM [33]. Hydrogen bonds between side amino acid residues of different subunits of protein complexes are highlighted in blue.

**Figure 2 biomolecules-12-00106-f002:**
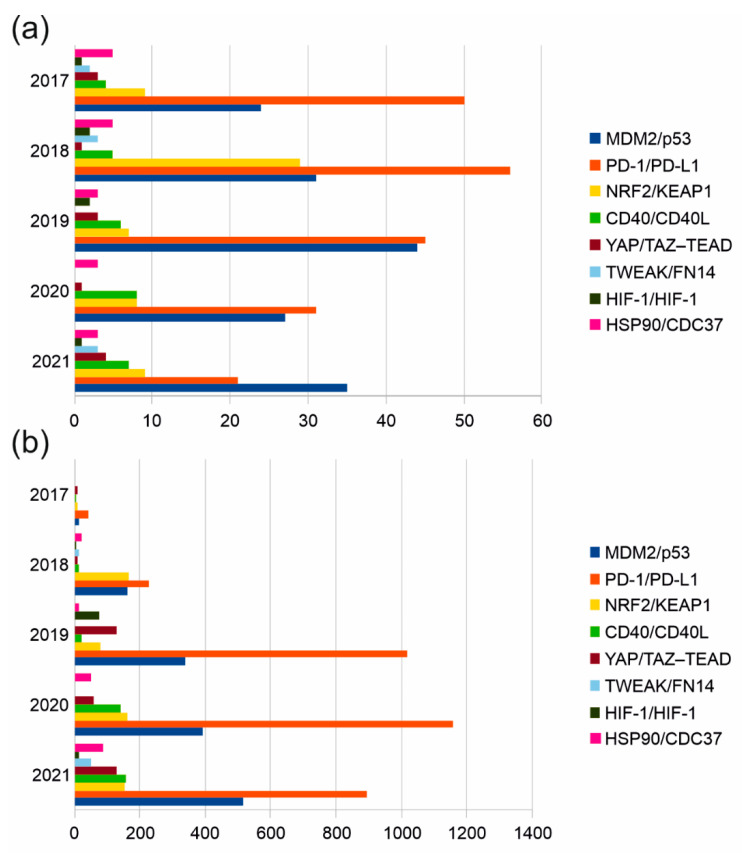
The number of publications indexed by Scopus and devoted to the pharmacological modulation of the binary complex formation of clinically significant proteins (**a**) and their citation rate (**b**).

**Figure 3 biomolecules-12-00106-f003:**
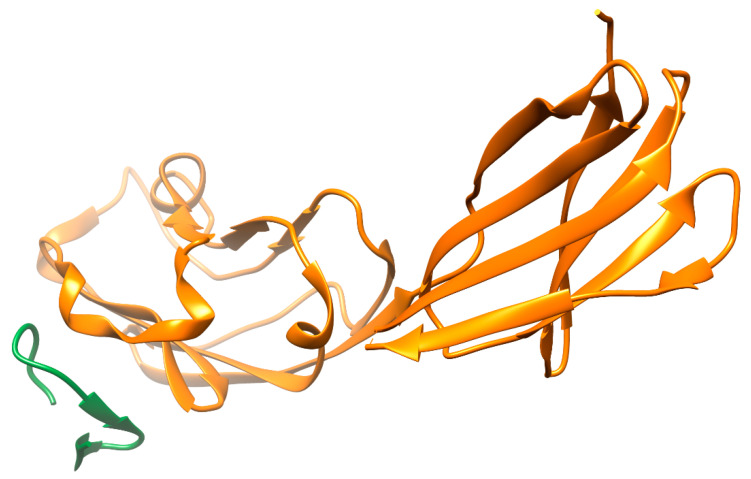
Crystallographic model of PD-L1 with interfacial peptide (PDB ID: 6pv9 [91]) using UCSF Chimera for visualization [98] and the main parameters of the interface calculated using PDBePISA: interface area—607.0 Å^2^, solvation free energy gain upon formation of the interface = (−10.7 kcal/mol), hydrogen bonds = 5, salt bridges = 2.

**Figure 4 biomolecules-12-00106-f004:**
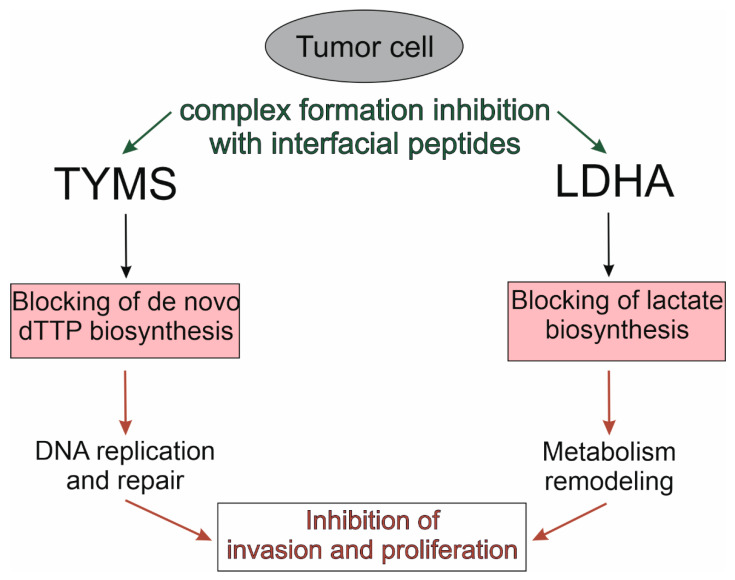
Pharmacological targeting of cancer-associated oligomeric metabolic enzymes: thymidylate synthase (TYMS) and lactate dehydrogenase A (LDHA).

**Table 1 biomolecules-12-00106-t001:** Current design concepts of interfacial peptides.

Design Concepts	References
**Alteration of linear peptide amino acid composition**	
Substitution of l- for d-amino acids	[51,52]
Inclusion of non-proteinogenic amino acids	[53]
**Macrocyclization (stapled peptides)**	
Stapling via a dithiocarbamate linker	[54]
Stapling via dialkynyl linker	[55]
Stapling via photoisomerizable diarylethene residue	[56]
**Peptides conjugates**	
Stapled peptide, conjugated with a cell-penetrating peptide	[57]
Lanthanum oxyfluoride nanoparticles with the affine peptide and monoclonal antibody	[58]
A conjugate of affine peptide and serum albumin	[59]
Self-assembling nanostructures based on lipopeptide conjugates	[60]
**Peptide grafting**	
Peptide grafting into ubiquitin scaffold	[61]
Peptide grafting into scaffold based on cyclic motif helix-loop-helix	[62]
Peptide grafting into scaffold based on N-terminal domain of chimeric oncoprotein Bcr/Abl	[63]
**Combined approaches**	
Alpha-helical cyclic peptide with inclusion of D- and N-methylated amino acids	[64]

**Table 2 biomolecules-12-00106-t002:** Median values of the main interfacial parameters of MDM2/peptide and MDM2/compound complexes.

Parameters/ Complexes	Interface Area of a Complex, Å ^2^	Percent of Protein SASA ^1^ Occupied by a PPI Inhibitor	SASA of an Inhibitor, Å ^2^	A Number of Hydrogen Bonds
Protein/peptide ^2^	560	8.3	1332	3
Protein/compound ^3^	389	5.1	720	1
Ratio	1.43	1.62	1.85	3

^1^ SASA—Solvent-Accessible Surface Area. ^2^ PDB ID: 4HFZ, 3EQY, 3JZS, 3FE7, 5UML, 3TPX, 1T4F, 2AXI, 6Y4Q, 3IWY, 7AD0, 5VK1, 5UMM, 5VK0, 4RXZ. ^3^ PDB ID: 1RV1, 1T4E, 2LZG, 3JZK, 3LBK, 3TU1, 3VZV, 3W69, 4DIJ, 4ERE, 4HG7, 6I3S, 6GGN, 5OAI, 4WT2.

**Table 3 biomolecules-12-00106-t003:** Functional enrichment analysis of a set of target proteins.

Subcategory Name (FDR B & Y Value)	Proteins	Enrichment Ratio, %
Protein domain specific binding (3.868 × 10^−4^)	Bax, Bcl-2, KEAP1, CRM1, Mdm2, NRF2, p53, Hsp90, CD40	36
Response to cytokine (1.680 × 10^−9^)	Fn14, TYMS, Cdc37, BIRC5, AKT, HIF-1, Bcl-2. KEAP1, NFAT5, TWEAK, YY1. NRF2. YAP, p53, Hsp90, CD40	64
Transcription regulator complex (2.621 × 10^−3^)	MTA1, NFAT5, YY1, RbAp48, NRF2, YAP, p53	24
Hematological neoplasm (8.940 × 10^−3^)	Bax, XIAP, AKT, Bcl-2, CRM1, Mdm2, TWEAK, NRF2, YAP, p53	40
Platinum drug resistance (3.236 × 10^−7^), Apoptosis (5.256 × 10^−7^) Pathways in cancer (3.750 × 10^−5^)	Bax, XIAP, BIRC5, AKT, Bcl-2, Mdm2, p53, Hsp90	32
Doxorubicin (6.303 × 10^−11^)	Fn14, TYMS, Bax, XIAP, BIRC5, PDCD1, AKT Bcl-2, KEAP1, Mdm2, LDH5, NFAT5, RbAp48, NRF2, YAP, p53, Hsp90, CD40	72
Triple Negative Breast Neoplasms (1.822 × 10^−13^)	Fn14, TYMS, Cdc37, Bax, XIAP, BIRC5, PDCD1, AKT, HIF-1, Bcl-2, MTA1, KEAP1, CRM1, Mdm2, LDH5, TWEAK, NRF2, YAP, p53, Hsp90	80
Diffuse Large B-Cell Lymphoma (9.284 × 10^−14^)	TYMS, Bax, XIAP, BIRC5, PDCD1, AKT, HIF-1 Bcl-2, MTA1, KEAP1, CRM1, Mdm2, YY1, NRF2, YAP, p53, Hsp90, CD40	72
Renal Cell Carcinoma (1.087 × 10^−12^)	Fn14, TYMS, Bax, XIAP, BIRC5, SRPK1, PDCD1, AKT, HIF-1, Bcl-2, KEAP1, CRM1, Mdm2, LDH5, TWEAK, YY1, NRF2, YAP, p53, CD40	80

## Data Availability

Not applicable.

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
