# Peer review of "Interfacial Peptides as Affinity Modulating Agents of Protein-Protein Interactions"

_biomolecules, 2022, doi:10.3390/biom12010106_

Round 1

Reviewer 1 Report

The review presented by Ershov and co-authors aims to update the current state-of-the-art in the field of interfacial peptides as modulating agents of protein-protein interactions (PPIs) related to several diseases, such as diabetes mellitus and Alzheimer's. In addition, the authors also listed several clinically significant PPIs from the number of publications and the level of their citation available in the literature over the past five years (Scopus and Google Scholar). Each of these clinically significant PPIs is presented in detail in the review. 

The review is well written, fairly well-referenced, and relevant for the field. That said, this study is interesting to the readers of the Biomolecules, but some small points must be addressed. Next, I present some points that deserve attention before acceptance.

1-There are some typos throughout the text (e.g., KD instead of KD, Å2 instead Å2).

2-On Pg. 5, lines 178-179, the PPI HSP90/CDC37 is listed twice. The same is true for the PPI BIRC5/CRM1 (on Pg. 5, line 183).

3-On Pg. 6, lines 202 and 233, I suggest replacing "homooligomeric'' with homo-oligomeric and "heterooligomeric'' with hetero-oligomeric, respectively.

4-On Pp. 6 and 9, lines 203 and 374, the PPIs LDH5 and SRPK1 are presented in detail but neither was mentioned at the beginning of section 4. Did this happen because of point 2 mentioned above? Otherwise, it would be good to mention them at the beginning of section 4.

5-It is known that inflamed tissues and tumors have a slightly more acidic environment than that found in healthy cells. Therefore, it is desirable that peptides are sensitive to changes in the pH of the medium and the presence of other charged entities. The authors could briefly mention the charge regulation process in section 3 (see, for example, Srivastava et al., Langmuir, 33, 2017; da Silva, Biophys. Rev., 9, 2017; de Oliveira et al., J. Chem. Theory Comput., 16, 2020). In addition, it has also been recently shown that the presence of charge patches along the polyelectrolyte is also an important factor in the stability of the interaction (see Huang et al., Macromolecules, 54, 2021; de Carvalho et al., Phys . Chem . Chem . Phys ., 23, 2021; Lunkad et al., Polymers, 13, 2021).

6-The colors used in Figure 2 make it difficult to understand the graph, especially Figure 2B. The colors that illustrate the years are very similar to each other. I suggest using other colors or using another way to represent the data.

Author Response

Reviewer 1.

Comments and Suggestions for Authors

The review presented by Ershov and co-authors aims to update the current state-of-the-art in the field of interfacial peptides as modulating agents of protein-protein interactions (PPIs) related to several diseases, such as diabetes mellitus and Alzheimer's. In addition, the authors also listed several clinically significant PPIs from the number of publications and the level of their citation available in the literature over the past five years (Scopus and Google Scholar). Each of these clinically significant PPIs is presented in detail in the review. 

The review is well written, fairly well-referenced, and relevant for the field. That said, this study is interesting to the readers of the Biomolecules, but some small points must be addressed. Next, I present some points that deserve attention before acceptance.

Authors: Thanks the Reviewer for careful reading and valuable remarks! 

1-There are some typos throughout the text (e.g., KD instead of KD, Å2 instead Å2).

Authors: Agree. It was corrected

2-On Pg. 5, lines 178-179, the PPI HSP90/CDC37 is listed twice. The same is true for the PPI BIRC5/CRM1 (on Pg. 5, line 183).

Authors: Agree. It was corrected

3-On Pg. 6, lines 202 and 233, I suggest replacing "homooligomeric'' with homo-oligomeric and "heterooligomeric'' with hetero-oligomeric, respectively.

Authors: Agree. It was corrected.

4-On Pp. 6 and 9, lines 203 and 374, the PPIs LDH5 and SRPK1 are presented in detail but neither was mentioned at the beginning of section 4. Did this happen because of point 2 mentioned above? Otherwise, it would be good to mention them at the beginning of section 4.

Authors: Agree. We have added the following text fragment in the section 4. 

It can also be focused on cancer-associated proteins such as LDH5 and SRPK1. Unlike LDH5, protein kinase SRPK1 is monomeric and phosphorylates a number of protein substrates with RS domains. Thus the contact area of stable protein complexes with the participation of modifying enzymes is another molecular target of interfacial peptides.”

5-It is known that inflamed tissues and tumors have a slightly more acidic environment than that found in healthy cells. Therefore, it is desirable that peptides are sensitive to changes in the pH of the medium and the presence of other charged entities. The authors could briefly mention the charge regulation process in section 3 (see, for example, Srivastava et al., Langmuir, 33, 2017; da Silva, Biophys. Rev., 9, 2017; de Oliveira et al., J. Chem. Theory Comput., 16, 2020). In addition, it has also been recently shown that the presence of charge patches along the polyelectrolyte is also an important factor in the stability of the interaction (see Huang et al., Macromolecules, 54, 2021; de Carvalho et al., Phys . Chem . Chem . Phys ., 23, 2021; Lunkad et al., Polymers, 13, 2021).

Authors: Agree. We have re-written the point concerning pH-sensitivity of peptides. Following text was added. 

Small molecule therapeutics can be fused with peptides that are sensitive to changes in the pH of the medium. This makes targeted delivery of peptides to tissue sites possible within a slightly more acid milieu (less than pH 7.0), which is the characteristic of tumors and/or their microenvironment [PMID: 32022813] as well as inflamed tissues [PMID: 28914424]. Under slight acid conditions, a pH-sensitive peptide takes the conformation of the alpha-helix that favors its incorporation into the cell membrane [66,67]. Srivastava and co-authors demonstrated that the charge regulation mechanism is the most important contributor in protein-polyelectrolyte complexation regardless of pH and other physical chemistry parameters using constant-pH Monte-Carlo simulations [PMID: 28859478]. In addition, it has also been recently shown that the presence of “charged patches” along the polyelectrolyte is also an important factor in the stability of the protein/peptide interactions [PMID: 34821240, PMID: 33435335].”

6-The colors used in Figure 2 make it difficult to understand the graph, especially Figure 2B. The colors that illustrate the years are very similar to each other. I suggest using other colors or using another way to represent the data.

Authors: Agree. The new version of the Figure 2 was created.

The figure legend was also corrected.

Figure 2. The number of publications indexed by Scopus and devoted to the pharmacological modulation of binary complex formation of clinically significant proteins (a), and their citation rate (b).”

Reviewer 2 Report

Minor revisions

Please add a figure with a docked peptide/cyclic peptide

Line 11    exchange „disturbing“ by „modulating“   , “disturbing” suggests that the complexes are no longer functional

Line 16      “In this review we update ….”

Line 18  “ Over the past yeas the scientific interest has been      “   skip in our opinion and the time scale

Line 28    exchange is by show

Line 30    and are disease-associated

Line 34 as one rule

Line 39-43    Perhaps:     To increase the pharmacological specificity of a PPI network, it is more appropriate to modulate peripheral clusters represented by molecular pathways containing functionally significant PPIs that are more abundant in pathology and predominantly absent in normal organisms.

Line 73    with lots of

Line 78   cell penetration capacity

Line 83- 87  this sentence  to the end of the introduction

Line 93  contact areas

Line 100-103  numbers into the brackets   10-3 , 10-9  ,10-12 M

Line 106  which later will also

Line 131   why you exclude the phi-phi staking of aromatic aa-rings?

Line 149  skip below

Line 151  this is not the whole story: cyclization can also be done by oxidation of cysteine containing peptides , or in the presence of catalytic metals

Line 160     add:   or solid tissues

Line 163  cell surface proteins

Line 165  This makes targeted delivery of peptides to tissue sites possible within an acid milieu (less than pH 7.0), which is the characteristic of tumors and their/or their microenvironment.

Line 169  incorporation does not mean forced internalization

Line 175   skip over the past 5 years 

Line 179  scientific studies  instead of publications

Skip sentence line 199-201

Line 206  which simulation program?

Line 245 KD  D subscript    (KD in pM range)

Line 253 KD  D subscript    Follows down this section 

Line 254 In silico modeling 

Line 263  a study by Lundsten . ref.79  addressed this problem

Line 269  skip with the help : instead   by using

Line 291  in vitro in vivo  follow down

Line 296  by?  Add author not only reference,  a review should be readable not only someone always had to look into the references

Line 378  skip residues

Chapter 4.2.8   and below:        could the numbers at the aa sequences be subscript

Add a cartoon /drawing( before or after the conclusion) showing some of the described target proteins in cellular and pharmacological context as relation as a therapeutical agent.

Perhaps for the conclusion:  the new in silico alphafold2 structure analysis of most soluble and membrane proteins (i.e. receptors) will help to further optimize binding properties of therapeutic peptides to their target proteins in and as well as outside the cell membranes. ….   You can add some more if you like. ;-)

Author Response

Reviewer 2

Comments and Suggestions for Authors

Authors: Thanks the Reviewer for careful reading and valuable suggestions!

Minor revisions

Please add a figure with a docked peptide/cyclic peptide

Authors: Agree. 

We have added Figure 3 to the subsection 4.2.2. “PD-1/PD-L1” that helps to visualize of the complex of PD-L1 with bound interfacial peptide as an example.

The following text was added.

Figure 3 shows a model of the PD-L1 complex with an interface peptide targeting the PD-1 contact area.

Figure 3. Crystallographic model of PD-L1 with interfacial peptide (PDB ID: 6pv9 [85]) using UCSF Chimera for visualisation [PMID: 15264254] and the main parameters of the interface calculated using PDBePISA: interface area - 607.0 Å2, solvation free energy gain upon formation of the interface = (- 10.7 kcal/mol), hydrogen bonds = 5, salt bridges = 2.

Line 11 exchange „disturbing“ by „modulating“, “disturbing” suggests that the complexes are no longer functional

Authors: Agree. It was corrected to more simplified and unambiguous text version: 

A phrase “PPI's modulation is aimed at both disrupting the complex assembly and/or destabilizing it.”

was exchanged by

Exogenous regulation of PPIs is aimed at both inhibiting the assembly and/or destabilization of protein complexes.

Line 16 “In this review we update ….”

Authors: Agree. It was corrected. 

In this review we update the current state-of-the-art in the field of interfacial peptides as potent modulators of a number of disease-related PPIs.”

Line 18 “ Over the past yeas the scientific interest has been “ skip in our opinion and the time scale

Authors: Agree. It was corrected.

Line 28 exchange is by show

Authors: Agree. It was corrected.

Line 30 and are disease-associated

Authors: Agree. It was corrected.

Line 34 as one rule

Authors: Agree. It was corrected.

Line 39-43 Perhaps: To increase the pharmacological specificity of a PPI network, it is more appropriate to modulate peripheral clusters represented by molecular pathways containing functionally significant PPIs that are more abundant in pathology and predominantly absent in normal organisms.

Authors: Agree. It was corrected.

Line 73 with lots of

Authors: Agree. It was corrected.

Line 78 cell penetration capacity

Authors: Agree. It was corrected.

Line 83- 87 this sentence to the end of the introduction

Authors: Agree. It was corrected.

Line 93 contact areas

Authors: Agree. It was corrected.

Line 100-103 numbers into the brackets 10-3, 10-9,10-12 M

Authors: Agree. It was corrected.

Line 106 which later will also

Authors: Agree. It was corrected.

Line 131 why you exclude the phi-phi staking of aromatic aa-rings?

Authors: The line was added with the phrase ”and pi-pi stacking interactions between aromatic amino acid rings [PMID: 33431014]”......

Thus, the presence of electrostatic complementarity between the surfaces of proteins allows the formation of salt bridges, which, in addition to hydrogen bonds and pi-pi stacking interactions between aromatic amino acid rings [PMID: 33431014], stabilize the protein-protein complex and increase its lifetime.”

Line 149 skip below

Authors: Agree. It was corrected.

Table 1 lists design concepts of interfacial peptides, modulating the clinically significant PPIs [50-63], which will be discussed.”

Line 151 this is not the whole story: cyclization can also be done by oxidation of cysteine containing peptides , or in the presence of catalytic metals

Authors: Agree. It was added in the text.

Сyclization can also be done by oxidation of cysteine containing peptides, or in the presence of catalytic metals.”

Line 160 add: or solid tissues

Authors: Agree. It was corrected.

…….which is especially important in the case of tumors or solid tissues. ”

Line 163 cell surface proteins

Authors: Agree. It was corrected.

These motifs are recognized by cell surface proteins….

Line 165 This makes targeted delivery of peptides to tissue sites possible within an acid milieu (less than pH 7.0), which is the characteristic of tumors and their/or their microenvironment.

Authors: Agree. It was corrected. 

Line 169 incorporation does not mean forced internalization

Authors: Agree. It was corrected. 

Under slight acid conditions, a pH-sensitive peptide takes the conformation of the alpha-helix that favors its incorporation into the cell membrane [66, 67]”

Line 175 skip over the past 5 years

Authors: Agree. It was corrected. 

We used Scopus literature database to search for publication statistics and citations (2017 - 2021).......”

Line 179 scientific studies instead of publications

Authors: Agree. It was corrected. 

We found that the greatest scientific interest (the number of scientific studies and the level of their citation)....”

Skip sentence line 199-201

Authors: Agree. It was corrected. 

The text fragment

Verification solves the problem of obtaining evidence of the modulating effect of peptides on the targeted protein complex. That can be measured in the simple system of highly purified protein preparations by assessing the binding kinetic (affinity) constants in the absence / presence of a peptide or in biochemical tests if applicable.”

was exchanged by 

Protein-peptide interactions are initially assessed through binding kinetic and affinity constants measurement or in biochemical tests.

Line 206 which simulation program?

Authors: Agree. We have added the following text in brackets:  

Using in silico simulations (FF14SB force field Amber 20 package and Rosetta) a peptide with a high affinity to the beta-sheet region…..“

Line 245 KD D subscript (KD in pM range)

Authors: Agree. It was corrected.

Line 253 KD D subscript Follows down this section

Authors: Agree. It was corrected.

Line 254 In silico modeling

Authors: Agree. It was corrected (in silico was highlighted with italic)

Line 263 a study by Lundsten . ref.79 addressed this problem

Authors: Agree. It was corrected. 

However, a study by Lundsten and co-authors [79] addressed this problem.

Line 269 skip with the help : instead by using

Authors: Agree. It was corrected.

Line 291 in vitro in vivo follow down

Authors: Agree. It was corrected. 

Line 296 by? Add author not only reference, a review should be readable not only someone always had to look into the references

Authors: Agree. It was corrected.

Line 378 skip residues

Authors. A term “amino acid residues” correspondes to a peptide or protein length

Chapter 4.2.8 and below: could the numbers at the aa sequences be subscript

Authors: Agree. It was corrected

Add a cartoon /drawing( before or after the conclusion) showing some of the described target proteins in cellular and pharmacological context as relation as a therapeutical agent.

Authors: Agree. The following text fragment was added at the beginning of the Conclusion section. 

For a panoramic view of the relationships between the cellular and pharmacological context, we performed a functional enrichment of a set of target proteins with the categories “gene ontology”, “human phenotype”, “pathway”, “drug” and “disease” using the ToppFun tool of the ToppGene Suite annotation platform [PMID: 19465376]. Table 3 shows the main subcategories containing the maximum number of target proteins. It follows from Table 3 that more than half of the target proteins in the gene-disease-drug axis are associated with the processes of neoplastic transformation, which, in principle, is logical, since this scientific field is one of the most important in ​​bio- and translational medicine. A strategy of pharmacological targeting of conservative contact regions of molecular complexes with participation of intensively studied PD-1, PD-L1, p53, Mdm2, Mdmx, KEAP1 and NRF2 proteins extrapolates to a group of homo-oligomeric metabolic enzymes, for example TYMS and LDHA, which function in homodimeric and homotetrameric forms, respectively, and overexpress in some tumors (Figure 4).”

Table 3. Functional enrichment analysis of a set of target proteins

Figure 4. Pharmacological targeting of cancer-associated oligomeric metabolic enzymes: thymidylate synthase (TYMS) and lactate dehydrogenase A (LDHA).

Perhaps for the conclusion: the new in silico alphafold2 structure analysis of most soluble and membrane proteins (i.e. receptors) will help to further optimize binding properties of therapeutic peptides to their target proteins in and as well as outside the cell membranes. ….You can add some more if you like. ;-)

Authors: Agree. We have added the following text fragment at the end of Conclusion section: 

Recently AlphaFold2 approach has emerged as a powerful in silico instrument for prediction of structure, topology, conformation and variants interpretation of most soluble and membrane proteins [PMID: 34074028, PMID: 34559429]. In the context of drug discovery it will contribute to optimization of binding models of pharmacological agents to their novel target proteins. However, predicted models of molecular complexes are still structural hypotheses requiring experimental verification.”
